# Relationship Between Facial Melasma and Ocular Photoaging Diseases

**DOI:** 10.3390/medsci13020061

**Published:** 2025-05-16

**Authors:** Lunla Udomwech, Chime Eden, Weeratian Tawanwongsri

**Affiliations:** 1Department of Ophthalmology, School of Medicine, Walailak University, Nakhon Si Thammarat 80160, Thailand; ludomwech@gmail.com; 2Division of Dermatology, Jigme Dorji Wangchuck National Referral Hospital (JDWNRH), Thimphu 11001, Bhutan; chime.eden17@gmail.com; 3Division of Dermatology, Department of Internal Medicine, School of Medicine, Walailak University, Nakhon Si Thammarat 80160, Thailand

**Keywords:** melanosis, melasma, ultraviolet rays, ocular diseases, photoaging

## Abstract

**Background/Objectives:** Facial melasma is a common, chronic, and relapsing hyperpigmentation disorder, affecting up to 40% of adult women in Southeast Asia. Although most cases are mild, the condition may have a considerable psychological impact. Ocular photoaging diseases are also common and have been increasingly recognized in aging populations exposed to chronic sunlight. Ultraviolet (UV) radiation is implicated in both melasma and ocular photoaging; however, their relationship remains unclear. **Methods:** This cross-sectional study investigated the association between facial melasma and UV-induced ocular conditions among 315 participants aged 30–80 years at Walailak University Hospital, Thailand. Facial melasma was diagnosed clinically and dermoscopically, with severity assessed using the modified Melasma Area Severity Index. Ophthalmological examinations evaluated UV-related ocular conditions, including pinguecula, pterygium, climatic droplet keratopathy, cataracts, and age-related macular degeneration. Logistic regression analyses were performed, adjusting for age, sex, and sun exposure. **Results:** Facial melasma was identified in 66.0% of participants (n = 208), and nuclear cataracts were significantly associated with melasma (adjusted odds ratio, 2.590; 95% confidence interval, 1.410–4.770; *p* = 0.002). Additionally, melasma severity correlated with nuclear cataract severity (ρ = 0.186, *p* = 0.001). Other ocular conditions were not significantly associated with melasma. **Conclusions:** These findings suggest a shared UV-related pathogenesis between facial melasma and nuclear cataracts. Sun protection measures, including regular sunscreen use, UV-blocking eyewear, and wide-brimmed hats, may help mitigate the risk of both conditions. Further multicenter studies are warranted to confirm these findings and explore the underlying mechanisms.

## 1. Introduction

Melasma is a prevalent acquired hyperpigmentation disorder that primarily affects women, particularly those with a dark skin tone. It is characterized by symmetrical brown to grayish-blue patches, most commonly appearing on sun-exposed areas of the face [1,2]. The prevalence of melasma varies significantly, ranging from 1 to 50%, depending on the study population [1]. In Southeast Asia, melasma affects approximately 40% of adult women and 20% of adult men [3]. Du et al. reported that the prevalence of melasma in Indonesia reached 100% among individuals aged 41–50 years [4]. The risk factors for melasma include hormonal changes, genetic predisposition, thyroid disorders, psychological stress, air pollution, and ultraviolet (UV) radiation [5,6,7,8]. Ultraviolet radiation can be categorized into three regions: UVA (315–400 nm), UVB (280–320 nm), and UVC (100–280 nm) [9]. Ultraviolet radiation, particularly UVA and UVB, is a key trigger of melasma, driving increased melanogenesis and contributing to hyperpigmentation. It causes deoxyribonucleic acid (DNA) damage and induces skin inflammation by stimulating melanogenic cytokines and growth factors that enhance melanin production. UV radiation can induce up to 10,000 DNA lesions per cell per day. In response, cells activate the DNA damage checkpoint, leading to cell cycle arrest to facilitate repair. If the damage is irreparable, programmed cell death (apoptosis) may occur to maintain genomic integrity [10]. In melasma, ultraviolet-induced oxidative stress exacerbates pigmentation by upregulating melanocyte-specific genes and enhancing melanin synthesis [11,12,13]. Ultraviolet exposure significantly contributes to the key pathomechanisms of melasma, including the inappropriate activation of melanocytes, aggregation of melanin and melanosomes in the dermis and epidermis, increased mast cell count along with solar elastosis, alterations in the basement membrane, and enhanced vascularization [14].

Ultraviolet radiation not only causes photochemical damage to cellular DNA, leading to photo-related conditions such as melasma, but also contributes to ocular photoaging diseases. The cornea absorbs a substantial amount of UV radiation, particularly UVB and UVC [15]. Chronic UV exposure induces genetic instability and conjunctival inflammation, thereby disrupting the tear film. Additionally, UV radiation triggers oxidative stress, resulting in the accumulation of reactive oxygen species (ROS) and mitochondrial apoptosis in corneal epithelial cells, which contribute to the development of various corneal lesions [16,17]. Exposure to UV-B is a recognized risk factor for lens abnormalities, as the lens absorbs most of the UV radiation that reaches it. The anterior capsule filters approximately 60% of this radiation; however, prolong exposure can induce oxidative stress, apoptosis in lens epithelial cells, and the formation of cortical opacities [18]. Although the retina receives minimal UV radiation, approximately 1% below 340 nm and 2% between 340–360 nm, it remains vulnerable to damage. Ultraviolet-A radiation can penetrate deeper into the eye, causing oxidative stress and potentially contributing to degenerative diseases [19]. Common eye conditions or diseases associated with UV radiation exposure include those affecting the conjunctiva and cornea (e.g., pinguecula, pterygium, and climatic droplet keratopathy), lens (e.g., cataracts), and retina (e.g., age-related macular degeneration) [20,21,22].

As melasma and the aforementioned eye conditions or diseases share pathophysiological mechanisms involving UV radiation exposure, they are often observed together in clinical practice. However, few studies have investigated the relationship between these conditions. Therefore, researchers have initiated studies to explore the association between facial melasma and UV radiation-related eye diseases.

## 2. Materials and Methods

### 2.1. Study Design and Participants

This cross-sectional study was conducted between March 2024 and February 2025 at the Walailak University Hospital in Thailand following approval by the Walailak University Ethics Committee (WUEC-24-055-01). Written informed consent was obtained from all participants after a comprehensive explanation of the study. This study adhered to the principles of the Declaration of Helsinki and the International Conference on Harmonization of Good Clinical Practice. This study was registered with the Thai Clinical Trials Registry (TCTR20240409006). Participants were recruited at a general practitioner clinic by a non-physician investigator to minimize undue bias. Eligible participants aged 30–80 years underwent thorough skin examinations by a dermatologist and comprehensive eye assessments by an ophthalmologist. Exclusion criteria included a history of bilateral ocular surgery or laser treatment, pregnancy or lactation, recent treatment for facial melasma within the past six months, and current use of retinoids or hormone therapy.

### 2.2. Data Collection

Baseline characteristics, including sex, age, occupation, history of ocular trauma or microtrauma, comorbidities, current medications, smoking and drinking habits, family history of melasma, and frequency of wearing sunglasses, were collected. Additionally, factors related to photoaging diseases and the Sun Exposure and Protection Index (SEPI) [23] used to estimate sun-related habits were recorded. The SEPI includes two parts: Sun Exposure Habits (Part I), with scores between 0–32 indicating increased UV risk exposure, and the Propensity to Increase Sun Protection (Part II), with scores between 0–20, with higher scores reflecting a lower likelihood of adopting sun protection measures. Melasma was identified by a dermatologist (W.T.) based on clinical manifestations and dermoscopic findings [2,24]. Melasma was assessed using the modified Melasma Area Severity Index (mMASI), with scores between 0–24 [25]. Facial photographs were captured for documentation. Each participant underwent a comprehensive ophthalmological examination using a slit-lamp biomicroscope (Haag-Streit BQ900, Berne, Switzerland). The typical clinical presentations of facial melasma and ocular photoaging diseases are summarized in Appendix A [1,2,26,27,28,29,30,31,32,33]. Severity of ocular conditions, including pinguecula, pterygium, climatic droplet keratopathy, nuclear cataract, cortical cataract, posterior subcapsular cataract, and age-related macular degeneration, was assessed and rated on scales between 0–2, 0–3, 0–3, 1–7, 0–5, 0–5, and 0–5, respectively [26,34,35,36,37]. The higher score from either eye was recorded for the analysis.

### 2.3. Sample Size Calculation

Sample size was calculated using the following formula to estimate an infinite population proportion in a two-tailed test:$$n=\frac{Z_{1-\frac{\alpha }{2}}^{2}\times p\times (1-p)}{d^{2}}$$

Based on the study by Viso et al. [38], the parameters used for the calculation were a population proportion (*p*) of 0.25, margin of error (*d*) of 0.05, significance level (*α*) of 0.05, and Z-value (Z_0.975_) of 1.96. Using these values, the initial calculation yielded the required sample size of 289 participants. The sample size was increased to 332 participants to account for a potential loss of 15% at follow-up. This adjustment ensured that the study achieved a 95% confidence level and 80% statistical power, corresponding to an effect size of 0.22.

### 2.4. Statistical Analysis

Descriptive statistics, including frequency, percentage, mean, median, standard deviation, and interquartile range (IQR), were calculated. Relationships between variables were analyzed using odds ratios (ORs) with 95% confidence intervals (CIs). Correlations were assessed using the Pearson or Spearman correlation coefficients, depending on data distribution. Multivariate logistic regression analysis, adjusted for age, sex, and duration of sun exposure, was used to evaluate the association between facial melasma and sunlight-related ocular diseases. Receiver operating characteristic (ROC) curve analysis was performed to assess the discriminatory ability of facial melasma in predicting nuclear cataracts. Normality and multicollinearity assumptions were evaluated prior to the analysis. All statistical analyses were conducted using R version 4.3.2, except for the ROC curve analysis, which was performed using SPSS version 18 (IBM Corp., Armonk, NY, USA). Statistical significance was set at *p* < 0.05.

## 3. Results

A total of 316 participants were recruited in this study. One participant was excluded because of an incomplete questionnaire, leaving 315 participants for the final analysis. The study population was predominantly female (n = 226, 71.7%). Median age was 64.0 years (IQR: 17.0) and median body mass index (BMI) was 24.2 (IQR: 4.67). Details of the participants’ characteristics are presented in Table 1. Information on common medications associated with melasma and ocular photoaging diseases was collected, with statins being the most frequently used (n = 126, 40.0%). None of the participants reported the use of hormones, steroids, anticonvulsants, prolonged non-steroidal anti-inflammatory drugs, prolonged doxycycline, amiodarone, or hydroxychloroquine. Most of the participants were government employees (n = 165, 52.3%), followed by those engaged in agriculture (n = 55, 17.5%). Unemployed individuals (n = 38, 12.1%), business owners (n = 20, 6.3%), and the remaining participants, categorized under other occupations (n = 37, 11.8%), were also represented.

We found that the participants presented with the following conditions: pinguecula (n = 240, 76.2%), pterygium (n = 117, 37.1%), nuclear cataract (n = 216, 68.6%), cortical cataract (n = 59, 18.7%), posterior subcapsular cataract (n = 19, 6.0%), and age-related macular degeneration (n = 11, 3.5%). No climatic droplet keratopathy was detected in any participant. Melasma was observed in 208 (66.0%) participants. Participants without facial melasma and ocular photoaging diseases accounted for 107 cases (34.0%). Median severity scores for ocular and dermatological conditions were as follows: pinguecula, 1.0 (IQR: 1.0); pterygium, 0.0 (IQR: 2.0); nuclear cataract, 2.0 (IQR: 3.0); cortical cataract, 0.0 (IQR: 0.0); posterior subcapsular cataract, 0.0 (IQR: 0.0); age-related macular degeneration, 0.0 (IQR: 0.0); and melasma, based on mMASI, 1.8 (IQR: 3.6).

Univariate logistic regression analysis revealed a significant association between melasma and nuclear cataracts (OR: 3.239, 95% CI: 1.966–5.336, *p* < 0.001), whereas no significant associations were found with other ocular photoaging diseases (Table 2).

After the univariate logistic regression analysis of the presence of nuclear cataracts (Table 3), several significant associations were identified. Age was inversely associated with the presence of nuclear cataract (OR, 0.035; 95% CI, 0.010–0.119; *p* < 0.001). Higher BMI (OR: 0.393, 95% CI: 0.086–1.800, *p* = 0.027), hypertension (OR: 1.958, 95% CI: 1.153–3.324, *p* = 0.013), dyslipidemia (OR: 2.354, 95% CI: 1.411–3.926, *p* = 0.001), statin use (OR: 2.257, 95% CI: 1.345–3.786, *p* = 0.002), and the presence of melasma (OR: 3.238, 95% CI: 1.964–5.338, *p* < 0.001) were also significantly associated with the presence of nuclear cataract. Univariate and multivariate logistic regression analyses for other ocular photoaging diseases, including pinguecula, pterygium, cortical cataracts, posterior subcapsular cataracts, and age-related macular degeneration, are shown in Appendix A. Univariate and multivariate logistic regression analyses for facial melasma are shown in Appendix A.

As shown in Table 4, the multivariate logistic regression analysis identified three significant associations with the presence of nuclear cataracts. Age was positively associated (adjusted OR, 1.080; 95% CI, 1.050–1.110; *p* < 0.001) with the presence of nuclear cataract, whereas the presence of melasma was significantly associated with the presence of nuclear cataract (adjusted OR, 2.590; 95% CI, 1.410–4.770; *p* = 0.002). Additionally, wearing sunglasses outdoors was inversely associated with the presence of nuclear cataract (adjusted OR, 0.519; 95% CI, 0.268–0.978; *p* = 0.046). In the univariate analysis, wearing sunglasses outdoors was not significantly associated with nuclear cataracts; however, it became significant in the multivariate model after adjusting for potential confounders. Additionally, the Spearman’s rank correlation coefficient revealed a significant positive association (ρ = 0.186, *p* = 0.001, n = 315). This finding suggests that higher melasma severity is associated with greater nuclear cataract severity. Although the correlation coefficient indicates a modest relationship, the statistical significance highlights a meaningful link between the two conditions.

The ROC analysis assessing facial melasma as a predictor of nuclear cataracts yielded an Area Under the Curve of 0.635 (95% CI, 0.578–0.693), indicating fair discriminatory ability (Figure 1). The model demonstrated a sensitivity of 74.54%, specificity of 52.53%, and Youden index of 0.27, indicating good detection capability but a moderate false-positive rate. The positive predictive value was 77.4%, while the negative predictive value was 48.6%.

## 4. Discussion

Facial melasma and ocular photoaging diseases share a common risk factor: sun exposure [8,20,21,22]. However, studies exploring the link between these factors are limited. This study aimed to evaluate the relationship between facial melasma and UV-related eye diseases. Our findings showed that individuals with facial melasma had a 2.6-fold increased risk of developing nuclear cataracts, even after adjusting for age, sex, sun exposure, and other potential factors. Wearing sunglasses outdoors was associated with a 48% lower risk of nuclear cataract. Among individuals with melasma, the sensitivity for detecting nuclear cataracts was approximately 75%, whereas the specificity was approximately 53%.

Nuclear cataracts typically develop between the ages of 40 and 50, with their prevalence increasing significantly with age [39,40,41]. Similarly, our study demonstrated that for every one-year increase in age, the chance of developing a nuclear cataract significantly increases by 8%. This can be explained by a multifactorial process involving biochemical changes in the lens, oxidative stress, lifestyle factors, systemic health conditions, and environmental influences [41,42,43]. Exposomes, which include all the environmental factors experienced over a lifetime, play a crucial role in the development and progression of various health conditions. In cataracts and melasma, UV radiation is a major factor that drives disease development and contributes to its progression [44,45,46]. The UV index in Thailand is classified as extreme, exceeding 11 [47]. Among various locations, Southern Thailand has the highest UV index, averaging approximately 14 and potentially peaking at 15, especially during summer [48,49]. Thailand experiences consistently high temperatures owing to its geographical position within the tropics, where solar radiation is intense year-round. During summer, temperatures can reach up to 44 °C, with a rising trend over time [50,51]. Miyashita et al. found that eyes with high cumulative ocular UV exposure had significantly higher odds of developing nuclear cataracts (odds ratio, 5.35), posterior subcapsular cataracts (1.87), and retinal detachment (1.35) [52]. Prolonged exposure to sunlight exceeding 5 h per day was associated with a substantial increase in cataract prevalence, with 67.8% of cases classified as nuclear cataracts, followed by cortical (27.1%), anterior subcapsular (3.5%), and posterior subcapsular (1.6%) cataracts [53]. Kinoshita et al. reported a strong correlation between the prevalence of nuclear cataracts and combined effects of UV exposure and cumulative equivalent minutes at 43 °C, derived from computed lens temperature, with an adjusted coefficient of determination of 0.933 (*p* < 0.0001). Among the contributing factors, cumulative heat stress accounts for 52% of nuclear cataract prevalence, exceeding the impact of UV radiation (31%), and a decline in lens repair capacity (17%) [54]. Similarly, Kodera et al. demonstrated that cumulative heat exposure showed a stronger correlation with nuclear cataract prevalence, with an adjusted coefficient of determination of 0.920 when solar radiation was included and 0.850 without solar radiation [55]. The lens is exposed to UV radiation, with UVB and UVA being the most significant contributors to cataract formation [18,56]. In addition to UV radiation, short-wavelength visible lights, such as violet (407 nm) and blue (463 nm) lights, have been shown to induce cataract formation in ex vivo studies using porcine lenses [57]. The scattering of light within the lens varies depends on wavelength, with shorter wavelengths scattering more centrally and longer wavelengths scattering more towards the periphery. Notably, studies have suggested that shorter wavelengths experience greater scattering in the central region of the lens, which aligns with the characteristics of nuclear cataracts, in which central lens opacity is a defining feature [58]. Ultraviolet radiation contributes to cataract formation by generating ROS, which oxidize lens proteins, accelerate apoptosis, and promote protein aggregation, leading to opacity [59,60,61]. Key molecular changes include tryptophan oxidation, tyrosine crosslinking, and the accumulation of advanced glycation end products, which drive protein crosslinking and lens yellowing [56]. Additionally, ROS induce DNA damage and epigenetic modifications that impair repair mechanisms. Furthermore, epigenetic alterations disrupt stress response genes, whereas mitochondrial dysfunction activates apoptotic pathways and accelerates cataract progression [61,62,63].

Melasma is a multifactorial pigmentation disorder, with UV radiation and visible light—particularly blue light—being key contributing factors [64,65]. Heat also plays a significant role in melasma pathogenesis. One study found a significant association (*p* = 0.003) and positive correlation between prolonged exposure to cooking or occupational heat and increased melasma severity [66]. Ultraviolet radiation induces melasma through multiple molecular mechanisms, including the upregulation of *MC1R*, *GDA*, and *NQO1*, which enhances melanogenesis, while downregulating *H19* and *PPARA*, thereby affecting p53 signaling and lipid metabolism. Additionally, 0*WIF-1* modulation affects the Wnt/β-catenin pathway, further contributing to the development of melasma. Histologically, UV radiation promotes melanin production via fibroblast and keratinocyte signaling, degrades the basement membrane through matrix metalloproteinase activation, induces pendulous melanocytes, and disrupts lipid metabolism by altering sterol regulatory element-binding proteins and peroxisome proliferator-activated receptor gamma, leading to impaired skin barrier function [11]. Moreover, increased cutaneous temperature can act as a stimulus for melasma development, similar to UV exposure [67]. Heat exposure promotes melanocyte differentiation and activates signaling pathways involved in melanogenesis, including the mitogen-activated protein kinase pathway, which is a key mediator of UVB- and stress-induced melanogenesis. Specifically, heat activates extracellular signal-regulated kinases and p38, both of which are essential for melanocyte function and melanin production. Notably, when combined with UVB radiation, heat significantly enhances tyrosinase activation and melanogenesis compared with UVB exposure alone [68]. These findings highlight the complex interplay between UV radiation and heat in the pathogenesis of melasma.

To our knowledge, this is one of the first studies to systematically explore this association, providing valuable epidemiological insight into the broader effects of chronic UV exposure and photoaging. Our study demonstrated the co-occurrence of facial melasma and nuclear cataracts, along with a significant association between their severities. These findings highlight the importance of protection against the sun. Sun protection is the cornerstone of melasma management, preventing exacerbation and supporting treatment. Educating patients about the significance of sun protection and promoting adherence through accessible and affordable options can improve outcomes. Regular application of broad-spectrum sunscreen (sun protection factor ≥ 30), wearing hats, and seeking shade further enhance protection [9,69]. Similarly, sun protection is essential in preventing nuclear cataracts by reducing UV exposure. Approximately 41% of participants did not wear sunglasses outdoors, highlighting the need to raise awareness and promote the use of protective eyewear, particularly in regions with high UV index. Wearing sunglasses that block UVA and UVB, using hats and protective clothing, and avoiding direct sunlight during peak hours can significantly lower the risk of cataracts and other UV-related eye conditions [18,70]. Importantly, our findings suggest that facial melasma may serve as a visible marker of chronic UV exposure, potentially reflecting underlying or developing ocular changes such as nuclear cataracts. Recognizing this relationship supports the idea of integrated screening approaches and reinforces the importance of collaborative preventive strategies between dermatology and ophthalmology. Early evaluation of nuclear cataracts in individuals with facial melasma may be beneficial. Early detection enables timely surgical intervention, prevents cataract progression, improves postoperative visual outcomes, and enhances the quality of life [71,72]. Additionally, early management of nuclear cataracts can reduce the risk of postoperative complications, such as cystoid macular edema and hemorrhagic pigment epithelial detachment, which are more common in advanced nuclear cataracts [73].

This study has several limitations. First, the single-center study design may limit the generalizability of the findings, as differences in melasma and nuclear cataract prevalence across regions with varying UV indices could affect the observed associations. Multicenter studies have provided a broader perspective, particularly for populations with higher sun exposure. Second, the dynamic nature of facial melasma may affect its correlation with ocular photoaging diseases [74]. Melasma severity fluctuates due to seasonal and environmental factors, indicating that a single time-point assessment may not fully capture its relationship with ocular conditions. A prospective year-round study could help address this limitation. Third, unmeasured confounding factors, such as dietary habits, genetic predisposition, and cumulative sun exposure, were not accounted for and may have influenced the observed associations. Future studies should incorporate these variables to enhance the robustness of the findings. Fourth, the limited statistical power for certain ocular conditions is a limitation. Although facial melasma was significantly associated with nuclear cataracts, no significant association was observed with other ocular diseases. A larger sample size may be required to reliably detect such associations. Fifth, this study focused on the association between facial melasma and ocular photoaging diseases rather than developing a predictive model. A scoring system was not created due to the limited sample size and the risk of overfitting. Future studies with larger cohorts should explore predictive modeling to assess its clinical utility in identifying nuclear cataract risk, particularly in individuals with facial melasma.

## 5. Conclusions

This study demonstrated a significant association between facial melasma and nuclear cataracts, suggesting that cumulative UV exposure contributes to the pathogenesis of both conditions. Individuals with facial melasma have a 2.6-fold higher risk of developing nuclear cataracts independent of age, sex, and sun exposure habits. Additionally, a significant correlation was observed between melasma severity and nuclear cataracts. Protective measures, such as regular sunscreen use, UV-blocking eyewear, and wearing a wide-brimmed hat, were associated with a lower risk of nuclear cataracts, reinforcing the importance of comprehensive sun protection in preventing photoaging-related diseases. These findings highlight the need for further multicenter studies to confirm this relationship and investigate potential shared pathophysiological mechanisms.

## Figures and Tables

**Figure 1 medsci-13-00061-f001:**
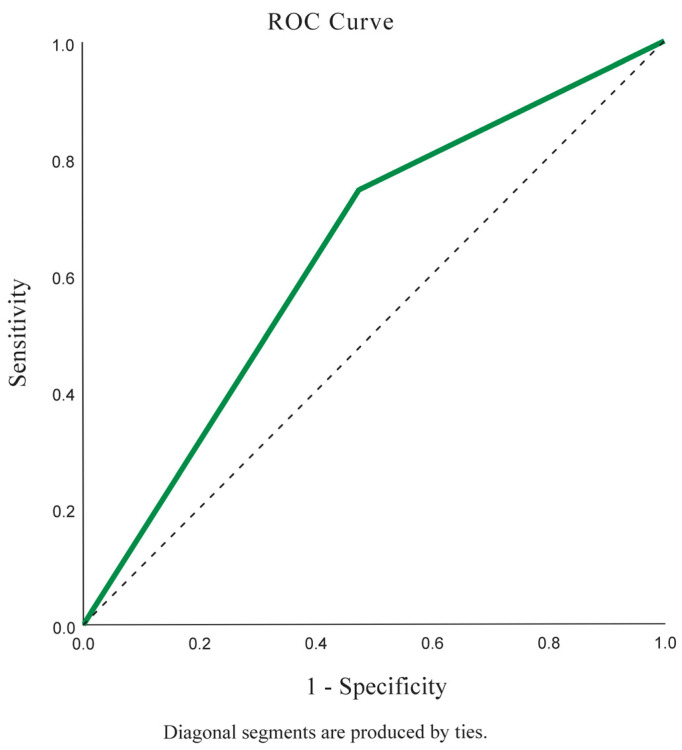
Receiver operating characteristic (ROC) curve for facial melasma as a predictor of nuclear cataracts.

**Table 1 medsci-13-00061-t001:** Demographic characteristics of the participants.

Characteristics	n (%)
Median age, years (IQR)	64.0 (17.0)
Female	226 (71.7)
Ethnicity	
Thai	315 (100.0)
Median BMI, kg/m^2^ (IQR)	24.2 (4.7)
Fitzpatrick skin type	
Type III	194 (61.6)
Type IV	101 (32.1)
Type V	20 (6.3)
Microtrauma	139 (44.1)
Hypertension	111 (35.2)
Diabetes mellitus	51 (16.2)
Dyslipidemia	132 (41.9)
Thyroid disorders	12 (3.8)
Dry eye	40 (12.7)
Related medications	
Statins	126 (40.0)
Anticonvulsants	1 (0.3)
Smoking	17 (5.4)
Alcohol consumption	29 (9.2)
Family history of melasma	118 (37.5)
Wearing sunglasses when outdoors	
Never	130 (41.3)
Rarely	20 (6.3)
Sometimes	60 (19.0)
Often	9 (2.9)
Always	96 (30.5)
Median SEPI Part I (IQR)	6.0 (6.0)
Median SEPI Part II (IQR)	4.0 (7.0)

Note: BMI, body mass index; IQR, interquartile range; kg/m^2^, kilograms per square meter; SEPI, Sun Exposure and Protection Index.

**Table 2 medsci-13-00061-t002:** Univariate logistic regression analysis of the association between facial melasma and ocular photoaging diseases.

Ocular Condition	B	OR	95% CI	*p* Value
Pinguecula	0.344	1.411	0.826–2.411	0.207
Pterygium	0.482	1.620	0.985–2.664	0.058
Nuclear cataract	1.175	3.239	1.966–5.336	<0.001
Cortical cataract	0.292	1.339	0.721–2.488	0.355
Posterior subcapsular cataract	0.115	1.122	0.414–3.041	0.821
Age-related macular degeneration	0.327	1.387	0.360–5.338	0.635

Note: B, regression coefficient; OR, odds ratio; CI, confidence interval.

**Table 3 medsci-13-00061-t003:** Univariate logistic regression analysis with respect to the presence of nuclear cataract.

Characteristics	B	OR	95% CI	*p* Value
Age (years)	−3.353	0.035	0.010–0.119	<0.001
Gender (female = 1, male = 2)	0.680	1.974	1.497–2.602	0.181
BMI (kg/m^2^)	−0.935	0.393	0.086–1.800	0.027
Fitzpatrick skin type				
Type III	NA	1.000	NA	NA
Type IV	0.414	1.513	0.891–2.568	0.126
Type V	1.140	3.127	0.885–11.048	0.077
Microtrauma (yes = 1, no = 0)	0.405	1.499	0.920–2.443	0.103
Hypertension (yes = 1, no = 0)	0.672	1.958	1.153–3.324	0.013
Diabetes mellitus (yes = 1, no = 0)	0.598	1.818	0.889–3.719	0.101
Dyslipidemia (yes = 1, no = 0)	0.856	2.354	1.411–3.926	0.001
Thyroid disorders (yes = 1, no = 0)	−0.091	0.913	0.268–3.108	0.885
Dry eye (yes = 1, no = 0)	0.077	1.080	0.524–2.226	0.835
Related medications				
Statins (yes = 1, no = 0)	0.814	2.257	1.345–3.786	0.002
Anticonvulsants (yes = 1, no = 0)	NA	NA	NA	NA
Smoking (yes = 1, no = 0)	0.419	1.520	0.483–4.786	0.474
Alcohol consumption (yes = 1, no = 0)	0.020	1.020	0.447–2.328	0.962
Family history of melasma (yes = 1, no = 0)	−0.429	0.651	0.400–1.059	0.084
Wearing sunglasses when outdoors (yes = 1, no = 0)	−0.237	0.789	0.484–1.285	0.342
Presence of melasma (yes = 1, no = 0)	1.175	3.238	1.964–5.338	<0.001
SEPI Part I	0.028	1.028	0.972–1.089	0.332
SEPI Part II	0.040	1.041	0.979–1.106	0.189

Note: B, regression coefficient; CI, confidence interval; NA, not applicable due to low sample size or as a reference category; OR, odds ratio; SEPI, the Sun Exposure and Protection Index.

**Table 4 medsci-13-00061-t004:** Multivariate logistic regression analysis with respect to the presence of nuclear cataract.

Characteristics	B	Adjusted OR	95% CI	*p* Value
Age (years)	0.074	1.080	1.050–1.110	<0.001
Gender (female = 1, male = 2)	0.718	2.050	0.888–5.010	0.102
BMI (kg/m^2^)	0.067	1.070	0.995–1.150	0.073
Fitzpatrick skin type				
Type III	NA	1.000	NA	NA
Type IV	−0.150	0.861	0.440–1.690	0.660
Type V	0.714	2.040	0.476–12.200	0.378
Microtrauma (yes = 1, no = 0)	0.205	1.230	0.681–2.230	0.496
Hypertension (yes = 1, no = 0)	−0.302	0.739	0.363–1.500	0.401
Diabetes mellitus (yes = 1, no = 0)	−0.086	0.918	0.407–2.180	0.840
Dyslipidemia (yes = 1, no = 0)	0.258	1.290	0.404–4.340	0.668
Thyroid disorders (yes = 1, no = 0)	−0.041	0.960	0.243–4.340	0.955
Dry eye (yes = 1, no = 0)	0.247	1.280	0.563–3.060	0.565
Related medications				
Statins (yes = 1, no = 0)	0.064	1.070	0.302–3.600	0.919
Anticonvulsants (yes = 1, no = 0)	NA	NA	NA	NA
Smoking (yes = 1, no = 0)	0.316	1.370	0.333–6.630	0.674
Alcohol consumption (yes = 1, no = 0)	−0.536	0.585	0.178–2.000	0.382
Family history of melasma (yes = 1, no = 0)	−0.207	0.813	0.442–1.500	0.506
Wearing sunglasses when outdoors (yes = 1, no = 0)	−0.655	0.519	0.268–0.978	0.046
Presence of melasma (yes = 1, no = 0)	0.951	2.590	1.410–4.770	0.002
SEPI Part I	−0.112	0.894	0.773–1.030	0.130
SEPI Part II	0.047	1.050	0.902–1.220	0.538

Note: B, regression coefficient; CI, confidence interval; NA, not applicable due to low sample size or as a reference category; OR, odds ratio; SEPI, the Sun Exposure and Protection Index.

## Data Availability

All data supporting the reported results are included in this article.

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
