# Peer review of "Relationship Between Facial Melasma and Ocular Photoaging Diseases"

_medsci, 2025, doi:10.3390/medsci13020061_

Round 1
Reviewer 1 Report
Comments and Suggestions for Authors
This present study investigated the association between facial melasma and UV-induced ocular conditions using logistic regression and ROC analysis. The authors reported that individuals with facial melasma have a 2.6 times higher risk of developing nuclear cataracts after adjusting for age, sex, sun exposure and other factors. Given that a large proportion of the participants did not wear sunglasses, the authors highlighted the need to increase awareness and promote UV protection for the eyes. I enjoyed reading the manuscript as it is well written. The methods are appropriate and well described. The authors have also adequately discussed the findings in the context of the current literature.
Author Response
Responses to reviewers
We sincerely appreciate your helpful comments and constructive suggestions, which have significantly improved the quality of this manuscript. We kindly request that the manuscript be considered for publication in Medical Sciences, a high-quality journal. We believe the manuscript is now greatly enhanced. As suggested, we have explained, point by point, the details of all manuscript revisions in our responses. Revisions corresponding to each reviewer’s comments are clearly highlighted in green in the revised manuscript. Our responses are presented below, with reviewer comments shown in italics.
Reviewer 1
Comment 1. This present study investigated the association between facial melasma and UV-induced ocular conditions using logistic regression and ROC analysis. The authors reported that individuals with facial melasma have a 2.6 times higher risk of developing nuclear cataracts after adjusting for age, sex, sun exposure and other factors. Given that a large proportion of the participants did not wear sunglasses, the authors highlighted the need to increase awareness and promote UV protection for the eyes. I enjoyed reading the manuscript as it is well written. The methods are appropriate and well described. The authors have also adequately discussed the findings in the context of the current literature
Reply
We sincerely appreciate your thoughtful review and positive feedback on our manuscript. We are pleased that you found the study well-written, the methods appropriate, and the discussion well-grounded in the current literature. Your observation regarding the lack of sunglass use among participants aligns with our key message about the importance of UV eye protection, especially in individuals with facial melasma. We agree that raising awareness about this issue is crucial for preventive healthcare, and we hope our findings contribute to this goal.
Reviewer 2 Report
Comments and Suggestions for Authors
I have highlighted a key comment along with a reference that must be included and addressed in the attached document.

Author Response
Responses to reviewers
We sincerely appreciate your helpful comments and constructive suggestions, which have significantly improved the quality of this manuscript. We kindly request that the manuscript be considered for publication in Medical Sciences, a high-quality journal. We believe the manuscript is now greatly enhanced. As suggested, we have explained, point by point, the details of all manuscript revisions in our responses. Revisions corresponding to each reviewer’s comments are clearly highlighted in green in the revised manuscript. Our responses are presented below, with reviewer comments shown in italics.
Reviewer 2
Comment 1 I have highlighted a key comment along with a reference that must be included and addressed in the attached document. The Article titled "Relationship between facial melasma and ocular photoaging diseases" investigates the association between facial melasma, a skin pigmentation disorder, and UV-induced ocular conditions, which are eye-related issues caused by ultraviolet radiation. This relationship is particularly important as both conditions are influenced by UV exposure, but their connection has not been clearly established before. The findings suggest a shared pathogenesis related to UV exposure between facial melasma and nuclear cataracts. The authors recommend sun protection measures, such as using sunscreen, UV-blocking eyewear, and wide-brimmed hats, to reduce the risk of both conditions. They also call for further multicenter studies to confirm these results and explore the underlying mechanisms. However, I have highlighted a few key comments for authors to address.
Reply
Thank you for your valuable feedback and for highlighting key comments. We have carefully addressed your comments in the revised manuscript (attached) and incorporated the suggested reference.
Comment 2 The clear impact of this study seems unclear and missing, which does not represent the importance of establishing the relationship between facial melasma and ocular photoaging diseases. Therefore, it is recommended to elaborate in the discussion section and highlight the authors’ view on it with appropriate references.
Reply
Thank you for your thoughtful feedback. We have revised the Discussion section to clearly highlight the importance and clinical relevance of establishing the relationship between facial melasma and ocular photoaging diseases. Specifically, we elaborated on how facial melasma may serve as a visible marker of chronic UV exposure, potentially reflecting subclinical ocular changes such as nuclear cataracts. This addition underscores the potential value of integrated preventive strategies and interdisciplinary screening approaches. The revised content, along with relevant references, can be found in the final paragraph of the Discussion section. (line 308–316)
Comment 3 Include the table that represents the characterization of both facial melasma and ocular photoaging diseases with appropriate references.
Reply
Thank you for your helpful suggestion. We have added a supplementary table 1 characterizing both facial melasma and ocular photoaging diseases, along with appropriate references. This has been incorporated into the revised manuscript and cited accordingly in the Methods section. (line 111–113)
Comment 4 Revise the abstract and include a brief about both diseases’ severity and prevalence, with relevant to this study and are missing.
Reply
Thank you for your valuable suggestion. We have revised the abstract to include a brief description of the severity and prevalence of both facial melasma and ocular photoaging diseases, as relevant to our study objectives. These additions were made while adhering to the word count limitation outlined in the journal's author guidelines. (line 12–16)
Comment 5 In the demographic table one, ethnicity’s information seems to disappear, and also make a separate tab or column, whichever way suits the authors to group, the demographic characteristic that eliminates the confusion in table 1.
Reply
Thank you for your kind observation. We would like to clarify that all participants in this study were of Thai ethnicity. We have adjusted the table format to improve clarity and reduce any potential confusion regarding the demographic characteristics.
Comment 6 This article also lacks a control group, which raises doubts when looking at the outcome of the relationship between both facial melasma and ocular photoaging diseases. So it is highly recommended that the author include a corresponding control group for each ocular condition and facial melasma.
Reply
Thank you for your thoughtful comment. We acknowledge the concern regarding the lack of a designated control group. As this was a cross-sectional study, participants without facial melasma and without ocular photoaging diseases (n = 107, 34.0%) served as the reference group in our logistic regression analyses. We have now clarified this point in the revised Results section. The odds ratios were calculated by comparing participants with each condition to those without, thus establishing a valid comparative framework within our study design. We appreciate your suggestion and hope this clarification adequately addresses the issue. (line 163–165)
Comment 7 A reference https://doi.org/10.3390/ijms20051073 must be included in the introduction section, page 2, lines 43 to 47, and wherever appropriate, which also well describes in detail the biological mechanisms of the DNA damage and stress in living cells.
Reply
Thank you for your helpful suggestion. We have now incorporated the recommended reference into the introduction section to support the discussion of the biological mechanisms underlying DNA damage and stress responses. This reference provides valuable context regarding the molecular pathways activated by ultraviolet-induced DNA lesions and their relevance to the pathogenesis of photoaging-related conditions. We believe this addition strengthens the scientific foundation of our manuscript. (line 50–53)
Comment 8 As the Cornea is exposed to the outer world environment and frequently encounters UV radiation, it regulates the various corneal pathophysiology and leads to blindness. Therefore, it is highly suggested to add the reference in the introduction section, page 2, lines 65 to 68, https://doi.org/10.3390/cells12212524, which details the diabetic and nondiabetic human cornea biological regulation mechanism.
Reply
Thank you very much for your thoughtful recommendation. We have carefully reviewed the suggested article [https://doi.org/10.3390/cells12212524], which provides valuable insights into the microRNA and protein cargos of limbal epithelial cell-derived exosomes and their regulatory roles in diabetic and non-diabetic corneal stromal cells. While this work significantly advances the understanding of diabetic corneal pathophysiology, its primary focus on cellular mechanisms in diabetic versus non-diabetic conditions diverges from the central theme of our study, which emphasizes ultraviolet-induced damage and its association with ocular photoaging conditions. Given this difference in scope, we respectfully believe that the inclusion of this reference would not directly enhance the narrative of our manuscript. Nonetheless, we appreciate the reviewer’s suggestion and the opportunity to consider relevant literature.
We look forward to hearing from you regarding our submission. We would be glad to respond to any further questions and comments that you may have.
Authors
Round 2
Reviewer 2 Report
Comments and Suggestions for Authors
As indicated in Comment 7, authors must update the bibliography to include the reference mentioned in Comment 7, which is currently absent from the manuscript. https://doi.org/10.3390/ijms20051073
Author Response
Responses to reviewers
We sincerely appreciate the reviewer’s insightful feedback, which has contributed to improving our manuscript. We have addressed the comment as detailed below. Our responses are presented below, with reviewer comments shown in italics.
Reviewer 2
As indicated in Comment 7, authors must update the bibliography to include the reference mentioned in Comment 7, which is currently absent from the manuscript. https://doi.org/10.3390/ijms20051073
Reply
Thank you for your valuable suggestion. We have now updated the bibliography to include the reference mentioned in Comment 7 (https://doi.org/10.3390/ijms20051073), as recommended.
We look forward to your response regarding our submission and remain at your disposal should any further clarification be required.
Kind regards,
The Authors